# Compact Slotted Waveguide Antenna Array Using Staircase Model of Tapered Dielectric-Inset Guide for Shipboard Marine Radar

**DOI:** 10.3390/s21144745

**Published:** 2021-07-12

**Authors:** Kyei Anim, Henry Abu Diawuo, Young-Bae Jung

**Affiliations:** Electronics Engineering Department, Hanbat National University, Daejeon 34158, Korea; kyeianim@gmail.com (K.A.); henrydiawuo@gmail.com (H.A.D.)

**Keywords:** beam scanning, dielectric-inset guide, high gain, low weight, marine radar, radiation efficiency, slotted waveguide array, X-band, size reduction, tapering

## Abstract

This paper presents a new configuration of a slotted waveguide antenna (SWA) array aimed at the X-band within the desired band of 9.38~9.44 GHz for shipboard marine radars. The SWA array, which typically consists of a slotted waveguide, a polarizing filter, and a metal reflector, is widely employed in marine radar applications. Nonetheless, conventional slot array designs are weighty, mechanically complex, and geometrically large to obtain high performances, such as gain. These features of the conventional SWA are undesirable for the shipboard marine radar, where the antenna rotates at high angular speed for the beam scanning mechanism. The proposed SWA array herein reduces the conventional design’s size by 62% using a tapered dielectric-inset guide structure. It shows high gain performance (up to 30 dB) and obtains improvements in radiation efficiency (up to 80% in the numerical simulations) and weight due to the use of loss and low-density dielectric material.

## 1. Introduction

Marine radars have been proven useful in commercial and military settings to provide information about other seacrafts and land targets [1] for diverse applications, such as coastal surveillance and surface target detection, collision avoidance, weather, and bird migration monitoring [2,3,4,5]. In this regard, X-band marine radars have become more attractive as they offer a high spatial and temporal resolution, low cost, flexibility, and installation ease [6]. The marine environment imposes severe constraints on antennas’ performance and mechanical design. Thus, the marine radar antenna needs to obtain fan-beam radiation patterns to operate successfully under these harsh conditions.

Several microstrip array antennas have been extensively employed in radar and other wireless communications systems [7,8,9,10,11,12,13,14,15,16,17]. Although they have promising features, such as low profile, low cost, and fabrication ease, their low-power handling capacity and high losses often make them less suitable for marine radar [18,19,20]. Consequently, the slotted waveguide array (SWA) antennas [21,22,23,24,25], based on the pioneering work published by Elliot [26,27,28], are widely employed in marine radar applications. This is because of their high-power handling capacity, low losses, and good phase stability. The conventional designs of SWA arrays typically consist of a slotted waveguide structure, a polarizing filter, and a large metal reflector to enhance directivity, especially in the vertical plane. By using this reflector device, especially in the lower frequency bands, the conventional SWA arrays become weighty, geometrically large, and mechanically complex [29]. They are, therefore, undesirable for marine radar systems of which the antenna rotates at high-angular speed. A miniaturization technique is thus required to make the traditional SWA more suitable for the marine environment without resorting to gain degradation. Nonetheless, reducing the reflector size as a simple miniaturization process results in considerable antenna gain reduction.

In this paper, a 62% size reduction of the SWA array was obtained by incorporating a tapered dielectric-inset guide structure in the conventional slot array design. At the same time, directivity in the vertical plane was improved significantly without the inclusion of the reflector device. The function of the dielectric structure with the staircase model is to smoothly transform the incident and strongly bound surface waves from the waveguide into free space characterized by minimum reflection and phase disparity. Thus, the results indicated that the proposed SWA array produces antenna gain up to 30 dB comparable to the conventional SWA designs. This new configuration of the SWA array was compared with its conventional counterpart to prove the practicality of the approach used in this paper.

## 2. Antenna Configuration

Figure 1 compares the geometry of the conventional SWA array, set as a benchmark here, to that of the proposed antenna based on the design specifications in Table 1. In both cases, the antenna consists of a slotted waveguide arrayed with ninety-four slot radiators to produce a fan-beam radiation pattern at the center frequency *f* = 9.41 GHz. The fan-beam, due to the long waveguide axial length (L) along the *y*-axis, has a narrow beamwidth of 1.2° in the horizontal (x) plane and a broader beamwidth of 20° in the vertical (x) plane to compensate for the roll of the ship [30]. The design, performances, and analysis of the antennas presented in this article have been performed using CST Microwave Studio software.

### 2.1. Slotted Waveguide Design

The slotted waveguide structure shown in Figure 2 consists of a WR-90 standard waveguide (a=22.86 mm, b=10.16 mm) with aluminum material and a wall thickness, t=1.27 mm. An array of tilted slots is milled into the narrow wall of a rectangular waveguide in order to generate the desired horizontally polarized electric fields [30], as specified in Table 1. The slots introduce discontinuities in the waveguide’s conducting walls to interrupt the flow of electric currents along the waveguide axial. Hence, each slot acts as a dual electric dipole, according to Babinet’s principle, to elicit radiations from the traveling waves propagating in the fundamental ${\mathrm{TE}}_{10}$ (transverse electric) mode in the waveguide [31]. The cutoff frequency for the ${\mathrm{TE}}_{10}$ mode is computed as $f_{c}=c/\left(2a\right)=6.56\;$GHz (c is the speed of light in a vacuum). The guided wavelength in this case is $\lambda_{g}=\lambda_{0}/\sqrt{1-{\left(f_{c}/f\right)}^{2}}=44.43$ mm ($\lambda_{0}$ is the free space wavelength).

The tilted slots wrap around the edges of the narrow wall and into the broad wall (see Figure 2). Thus, each slot has the slot tilt angle θ and slot depth δ parameters. By adjusting θ, the excitation strength from each slot can be controlled independently. The excitation is small for a large θ and zeroes for θ=0° which may be used in a non-uniformly excited array design with different *θ* values, as depicted in Figure 2, to decrease the sidelobe levels (SLLs). To reduce the number of design parameters, all slots have the same width *w* = 3 mm. The slot length for each slot in this case is $L_{r}=\left(b+t\right)/\cos\theta +2\left(\delta -t/2\right)=0.4625\lambda_{g}\;$. The set of values for δ is centered around 1.73 mm at resonance after optimization.

In the design of a non-uniformly SWA array in Figure 2, each slot’s dimensions are computed and optimized independently to attain the desired amplitude coefficient $a_{n}$ at resonance. Once$\;a_{n}\;$for each slot is set, the array factor (AF) based on Taylor distribution is estimated using (1) and (2) [32].
(1)$${\left(AF\right)}_{2M}=\sum_{n=1}^{M}a_{n}\cos\left[\left(2n-1\right)u\right]$$
(2)u=πd/λcosθ

*M* is an integer, *n* is the number of slots, and *d* is the slot spacing (see Figure 2). In this case, µ=0.5π/cosθ is used for the chosen slot spacing$\;d=0.5\lambda_{g}\;\left(i.e.,\;\lambda =\lambda_{g}\right)$. The AF is thus dependent on amplitude coefficient $a_{n}$ and the slot tilt angle θ of each slot to achieve the desired distribution to result in the radiation pattern computed as follows:(3)$${\bar{E}}_{total}={\bar{E}}_{single}\cdot \left(AF\right)$$

Since the slotted waveguide structure in Figure 2 is fed at the excitation port and terminated in a matched load (absorber) to achieve a traveling wave antenna, its equivalent circuit becomes a parallel conductance *G* at resonance, whereas the reactive susceptance *B* cancels out. Hence, the resonant conductance $g_{n}$ of *N* slot is computed from the amplitude coefficient $a_{n}$ using (4), while the slot tilt angle $\theta_{n}\;$is related to conductance $g_{n}\;$by (5). Once the initial θ values are set, the slot tilt angle is further optimized to obtain the final range of values from−10° to 15°.
(4)$$g_{n}=\frac{a_{n}^{2}}{\sum_{i=1}^{N}a_{i}^{2}};n=1,2,\ldots ,N$$
(5)$$\theta_{n}=-900.7g_{n}^{2}+237.6g_{n}+2$$

The main beam, which is perpendicular to the waveguide axis, lies in the forward direction along the *x*-axis (see Figure 1) because of the slot spacing $d=0.5\lambda_{g}$ and opposite tilt angles among adjacent slots. Thus, the co-polarization due to the desired horizontally polarized electric fields $E_{H}$ (see Figure 2) is superimposed in the phase, whereas the vertically polarized electric fields $E_{V}$, which contribute to undesired cross-polarization, among the adjacent slots, cancel out because they assume an equal amplitude but opposite direction. This phenomenon, in theory, inherently reduces the sidelobe levels of a waveguide slot array due to the low levels of cross-polarized signals.

### 2.2. Polarization Filter

The instantaneous electric fields *E* from the inclined slots decompose into the horizontal $E_{H}$ and vertical $E_{V}$ fields, as depicted in Figure 2. As mentioned earlier, the vertical electric fields $E_{V}\;$among the adjacent slots cancel out as the adjacent slots have equal and opposite tilt angles. However, in practice, the slot tilt angle θ among the adjacent slots varies for optimization and amplitude distribution purposes. In addition, due to fabrication errors, θ may differ from the numerical value in the simulation. Hence, the$\;E_{V}\;$among the adjacent slots do not cancel out entirely due to the varying phases and magnitudes. Therefore, high sidelobes are observed in the horizontal (xy) plane due to the measurable levels of unwanted cross-polarized signals (see Figure 3a).

In order to significantly reduce the sidelobes, a polarization filter is positioned in front of the slotted waveguide, as shown in Figure 2. The filter constitutes equispaced orthogonal walls with a spacing parameter $\psi \;\leq \;0.5\lambda_{g}$ to ensure adequate suppression of cross-polarized signals. The filter is positioned *ρ* distance away from the waveguide wall (see Figure 2). In principle, the effectiveness of the filter is inversely proportional to wall spacing ψ and proportional to the distance ρ between the filter and the waveguide wall. However, the attenuation of the $E_{V}\;$denoted by γ is determined primarily from the wall spacing ψ as expressed in (6) [33]. Referring to Figure 2, the filter’s orthogonal walls are positioned in such a way that they stand in between the adjacent slots to limit the efficient propagation of undesired *E_V_* to guarantee the sidelobes’ suppression, as observed in Figure 3a.
(6)$$\gamma =\frac{5.46}{2\cdot \psi }\cdot \sqrt{1-{\left(\frac{2\cdot \psi }{\lambda }\right)}^{2}}\left[\mathrm{dB}/m\right]$$

It should be noted that both the ψ and ρ parameters have an influence on the radiating slot and the radiation. Therefore, they are optimized and fixed as ψ=6 mm and ρ=20 mm.

### 2.3. Reflector

Referring to Figure 1, the conventional slot array design has a reflector device with a flare-out angle to increase the antenna’s aperture to increase the gain from 24 dB to 30 dB, as shown in Figure 3b, to satisfy the requirement in Table 1. Thus, the conventional SWA array has a dimension of 1920 mm × 165 mm × 84.3 mm (L × W × H). However, the SWA arrays with the reflector device are large in the antenna’s cross-section (i.e., yz-plane), mechanically complex, and weighty to rotate at a very high angular speed. The proposed antenna in Figure 1b is the result of the effort to design a reduced size SWA array by using a dielectric-inset structure, over which the bound surface waves are excited. Thus, its vertical size is about half that of the conventional case (see Figure 1a), giving the same vertical (xz) plane beamwidth, and shows improvement in weight and mechanical simplicity. After the full analysis of the SWA arrays in Figure 1 was performed, the following are the optimized design parameters: a=22.86, b=10.16 mm for WR-90, L=1920 mm, t=1.27 mm, θ=−10° ~ 15°, δ=1.73 mm, w=3 mm, ψ=6 mm, and ρ=20 mm.

## 3. Design and Analysis of the Proposed Slotted Waveguide Array Antenna

### 3.1. Evolution of the Proposed Case

To facilitate the development of the proposed antenna while still providing a physical understanding of its geometrical structure, the three-dimensional SWA arrays in Figure 1 are mirrored in their two-dimensional analogues in the xz-plane, as shown in Figure 4. Thus, the evolution of the proposed case is demonstrated in Figure 4. The conventional antenna, Type A, with a wide flare-angle reflector gives an ideal gain of more than 30 dB in Figure 5 at the expense of large antenna size. The simplest approach to reduce the antenna’s physical size is by reducing the reflector size. Unfortunately, this approach results in a significant decrease in antenna gain.

The solution to this problem was accomplished by incorporating a tapered dielectric-inset guide structure in the SWA array to produce antenna Type B in Figure 4. This approach immediately reduces the height of the conventional antenna by about 50%. However, in the Type B antenna configuration, an optimum taper profile can only be achieved when the length parameter $S_{x}$ in the forward direction is very long, up to 2.5λ, which undermines the antenna’s compactness. The relationship between $S_{x}$ and the gain of the antenna can be expressed mathematically in (7) [34].
(7)$$\mathrm{Gain}=8S_{x}/\lambda$$

Thus, the incident, strongly bound surface wave field from the slotted waveguide, which is butted to the base of the dielectric structure, is transformed smoothly into a radiation field that is characterized by maximum antenna gain (see Figure 5, Type B) with an in-phase field.

By truncating the long tapered section of the dielectric structure to produce antenna Type C (see Figure 4), the width of the antenna is reduced to 108 mm. Thus, a 29% size reduction is achieved in the forward direction as $S_{x}$ decreases to about 1.7λ. However, the gain of antenna Type C drops considerably to 26 dB in Figure 5a, since the phase disparity of the incident surface wave on the uniform cross-section of the truncated dielectric structure increases significantly. This causes the destructive interference of the surface wave field, transforming into a radiation field at the termination to result in a beam split (see Figure 5b, Type C).

To solve the beam splitting problem of antenna Type C without resorting to an increase in the antenna’s size, the dielectric-inset guide structure was modified by tapering the termination with a staircase approximation to form the end-gradient in antenna Type D. The staircase model of the tapered dielectric is synthesized as a series of short slab segments, wherein the uniform cross-sectional areas in the forward direction are diminished progressively, as shown in Figure 4. Thus, the width and height parameters of each dielectric slab segment of the staircase model of the taper profile were optimized individually to allow for the redistribution of propagating signals in the dielectric medium so that at the dielectric–air interface, all the signals exiting the medium assume the same approximated phase velocity to maximize radiation efficiency. In effect, a conformal beam pattern is restored for the proposed SWA array Type D as the gain improves to about 29 dB in Figure 5. By incorporating the dielectric-inset guide structure into the SWA design, the proposed antenna, Type D, achieves a 62% size reduction while maintaining the antenna’s high gain result, which is comparable to the conventional case-Type A.

It should be noted that the long axial length (L) of the antenna is preserved in all the designs to ensure that the beam has a narrow width in the horizontal plane to satisfy the required half power beamwidth (HPBW) of 1.2°.

### 3.2. End Gradient Optimization

The optimization of the dielectric-inset guide structure significantly improves the radiation pattern of the antenna. The taper profile of the optimized dielectric structure is depicted in Figure 6, wherein the end gradient of the dielectric is synthesized as a series of dielectric slab segments of equal width ∆w and uniform cross-sectional areas of gradually smaller heights $H_{i},\;i=1,\;2,\;\ldots ,6$. In other words, the tapered dielectric region is modeled as a sequence of sufficiently short (in terms of wavelength), uniform slab segments of diminishing cross-section in the forward direction, i.e., along the *x*-axis, as shown in Figure 6. The taper is segmented into five regions (i=2,3,…, 6) with successively smaller heights. In this step-synthesis technique, the step discontinuity on the dielectric structure is regarded as a radiating aperture of a serial uniform slabs. At each step discontinuity, the fundamental transverse electric (TE) surface wave, which is assumed to be incident in the +x direction from x=−∞, is perturbed to cause redistribution of the instantaneous phases of the surface wave field on the uniform cross-section for phase matching. Consequently, the optimum taper profile of the dielectric is taken as one which smoothly transforms the strongly bound surface wave field into a radiation field with minimal phase disparity and reflection. This suggests that surface wave fields constructively interfere as they assume the same approximated phase velocity to maximize radiation efficiency.

The instantaneous phase of the radiation field from the dielectric surface with no taper profile received at points A, B, and C, assumed to be within the far-field region from the antenna, i.e., $R=2D^{2}/\lambda$ (maximum linear dimension of an antenna is *D*), is shown in Figure 7a. It can be noted that there is a significant phase difference between the received fields at these points, resulting in a scattering wave to cause reflection and destructive interference. However, the introduction of the taper region of the optimized dielectric structure modeled with staircase approximation, in Figure 6, leads to a substantial reduction in the phase disparity at these received points within the desired band, as shown in Figure 7b. The radiation field at these points, therefore, assumes the same approximated phase velocity to mimic the behavior of a plane wave that maximizes radiation efficiency due to the constructive interference. The optimal taper profile of the dielectric in Figure 6 with the staircase model has geometrical parameters fixed as follows: $\Delta w=4\;\mathrm{mm},\;\;H_{2}=39.48\;\mathrm{mm},\;H_{3}=36.33\;\mathrm{mm},\;\;H_{4}=31.07\;\mathrm{mm},\;H_{5}=26.64\;\mathrm{mm}\;\mathrm{and}\;H_{6}=18.45\;\mathrm{mm}.$

It should also be noted that the base of the dielectric-inset guide structure, where the slotted waveguide is butted, has also been tapered using the staircase approximation (see Figure 6) to result in a minimum reflection and gradually transition the incident, strongly bound field to the dielectric medium. This feed taper, therefore, significantly reduces the Voltage Standing Wave Ratio (VSWR) on the uniform section of the guide to improve the radiation characteristics of the antenna. The matching slab segments at the base of the dielectric structure in Figure 6 have been optimized and fixed as: $\Delta w_{a}=2\;\mathrm{mm},\;\Delta w_{b}=3\;\mathrm{mm},\;\Delta w_{c}=5\;\mathrm{mm},\;H_{a}=10\;\mathrm{mm},\;H_{b}=20\;\mathrm{mm},\;\mathrm{and}\;H_{c}=35\;\mathrm{mm}.$

### 3.3. Dielectric Material Selection

The selection of a proper dielectric material for the design of the antenna’s dielectric structure is primarily based on the material’s relative permittivity $\left(\varepsilon_{r}\right)$, loss tangent (tanδ), and most importantly, weight or density. It is evident that as $\varepsilon_{r}\;$increases, the bandwidth reduces, and the radiation efficiency decreases due to the increase in the quality factor (Q-factor). It should be noted that as the $\varepsilon_{r}$ increases, the dielectric component no longer works under the TE_10_ mode required for a low sidelobe and high gain. This suggests that the lower the $\varepsilon_{r}$, the higher the radiation power for a fixed tanδ, as shown in Figure 8. Furthermore, extremely lightweight materials are highly recommended for the design of a dielectric component, since heavy materials undermine the compactness achieved by the proposed SWA array. Thus, to choose the right material for the dielectric component, several property data among commercial dielectric materials are compared in Table 2. It should be noted that model D has the lightest weight in terms of density and the lowest $\varepsilon_{r}$ among the materials. Therefore, it is chosen as the material of the dielectric component of the proposed antenna.

## 4. Results and Discussion

Figure 9a,b show a photograph of the fabricated SWA array and outdoor antenna installation with a radome, respectively.

The experimental results of the reflection coefficients shown in Figure 10a exhibit a broad bandwidth for $S_{11}<-20$ dB to cover the entire frequency band of interest. The experimental results of the gain and radiation efficiency are shown in Figure 10b. The measured gain curve, which ironically slightly outperforms the simulated one, has a peak gain of 30.06 dB at 9.41 GHz. The radiation efficiency is above 85% in the desired band of 9.38−9.44 GHz.

Figure 11a–f show the measured and simulated radiation patterns of the proposed SWA array in the xy-plane (horizontal) and xz-plane (vertical) at 9.38, 9.41, and 9.44 GHz. They show that measured beam patterns, to a large extent, agree well with the simulated results. However, the measured xz-plane beam patterns have SLL lower than −12 dB, which is substantially higher than that of the simulated results at the frequency points. This is attributable to the complexity of the antenna’s configuration. Thus, the combined effect of the different structural layers of the antenna has a higher tendency for errors to occur during manufacturing and assembly. This could easily be corrected when the antenna is assembled properly. The measured xy-plane HPBWs are all 1.2° at these three frequency points. The corresponding measured xz-plane HPBWs are 28°, 27.8°and 27.7°.

As the proposed SWA array operates in X-band, the conventional SWA array, set as a benchmark, should also operate in X-band for comparative physical size to ensure a fair comparison. Under this condition, an SWA array with a high-profile reflector for a high gain result was simulated. The two-dimensional equivalents in the xz  or vertical plane of the proposed and conventional models are shown in Figure 12a. The simulated gain curves are plotted in Figure 12b.
Ant-A: Proposed SWA array comprising of a slotted waveguide, compact dielectric structure, and miniaturized reflector.Ant-B: Commercial SWA array composed of a slotted waveguide and a relatively larger reflector.

The proposed antenna possesses an average gain of 29 dB, which has been found to be comparable to that of the conventional type. However, the proposed antenna configuration achieves a 50% size reduction in lateral size that lies along the z-axis and 29% in the forward or radiation direction along the x-axis. Thus, the antenna’s total size is reduced by 62%, while keeping its high gain within the band of interest. Moreover, a low loss, low-density material was chosen for the design of the dielectric structure to maintain this physical advantage over the conventional counterpart without resulting in a heavy weight and low efficiency.

Table 3 compares the requirements of the marine radar to the measurement data of the proposed SWA array. It shows that the proposed SWA array can fulfil all of the requirements except for the SLL in the vertical plane. This is primarily due to fabrication and assembly errors, which may result from misalignment between the different structural components of the proposed SWA array. Thus, proper assembly of the various components will easily alleviate the high SLL in the measured results.

## 5. Conclusions

In summary, a new configuration of slotted waveguide array (SWA) antennas, comprising a slotted waveguide, polarizing filter, and tapered dielectric-inset guide structure, was proposed for use in a shipboard marine radar in X-band. The concept of using the dielectric-inset guide structure to reduce the size of an SWA array was presented. Comparison between the proposed and the conventional SWA arrays was also presented, confirming the practicality of this approach. The experimental results indicated that a size reduction of 62% could easily be achieved for the SWA array, while preserving the antenna’s superior performance, i.e., a higher gain and radiation efficiency at the same time. A staircase model of the tapered section of the implemented dielectric structure was introduced. The incident, guided surface wave field on the uniform cross-section was perturbed for phase matching purposes to ensure that it transformed into a radiation field that is characterized by the maximum intensity with little reflection. Thus, the antenna exhibited a peak gain of 30.06 dB and radiation efficiency above 80% in the desired band of 9.38~9.44 GHz. The measured horizontal plane HPBWs are about 1.2° at the various frequency points. The corresponding measured xz-plane HPBWs are about 28°. The selected dielectric has low-density properties to maintain low weight. This proposed SWA array has promising physical and electrical features, making it more suitable for modern marine radar applications than the conventional case.

## Figures and Tables

**Figure 1 sensors-21-04745-f001:**
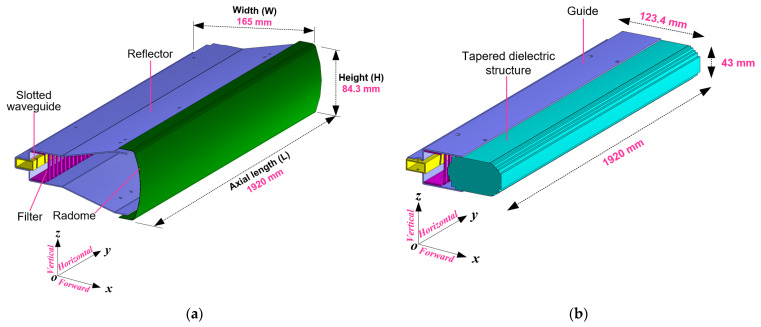
Full 3D views of slotted waveguide array antenna. (**a**) Conventional case; (**b**) Proposed case.

**Figure 2 sensors-21-04745-f002:**
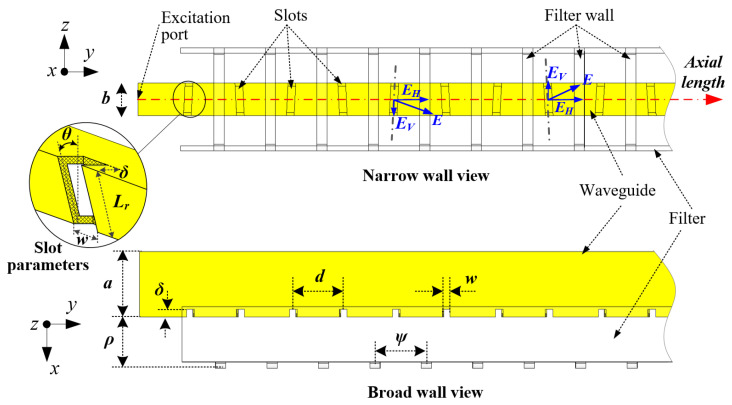
Configurations of the slotted waveguide antenna and the polarization filter used in the design of the SWA arrays.

**Figure 3 sensors-21-04745-f003:**
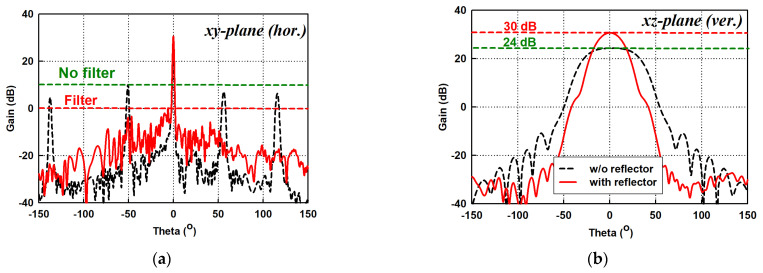
Radiation patterns of conventional SWA array. (**a**) Horizontal (xy) plane; (**b**) Vertical (xz) plane.

**Figure 4 sensors-21-04745-f004:**
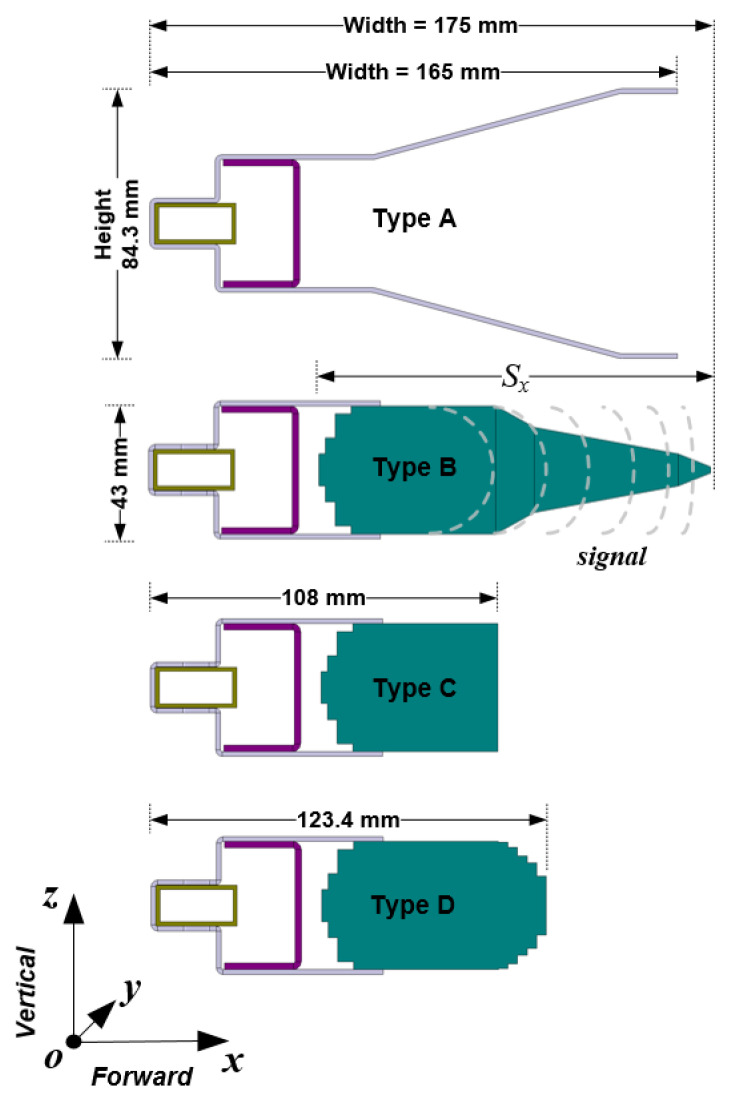
Two-dimensional analogue of four different SWA array models in the *xz* (vertical) plane.

**Figure 5 sensors-21-04745-f005:**
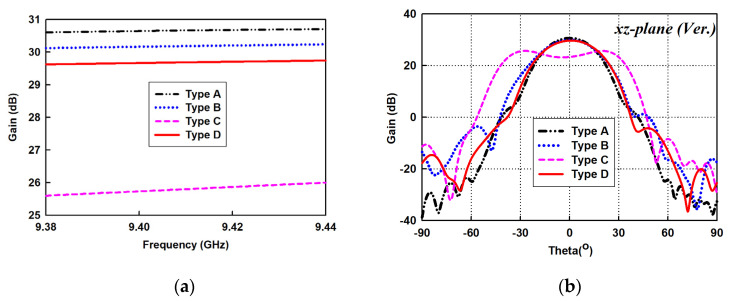
Simulated results for different SWA array models. (**a**) Gain; (**b**) Patterns in vertical (*xz*) plane.

**Figure 6 sensors-21-04745-f006:**
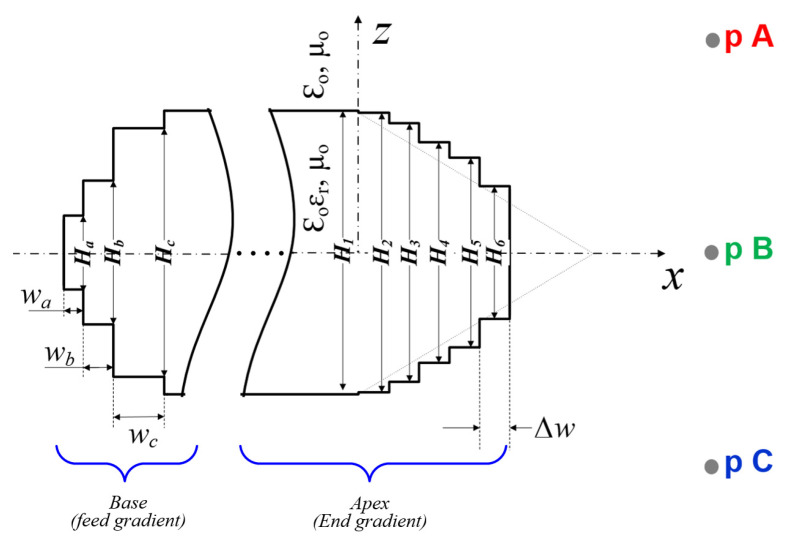
Geometry of the staircase model of the tapered dielectric-inset guide structure depicting the end gradient and feed gradient.

**Figure 7 sensors-21-04745-f007:**
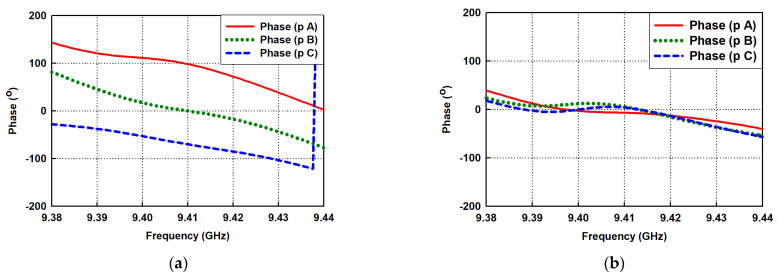
Simulated phase responses at three different points from the antenna’s dielectric rod structure with (**a**) No taper profile; (**b**) Taper region modeled with the staircase approximation.

**Figure 8 sensors-21-04745-f008:**
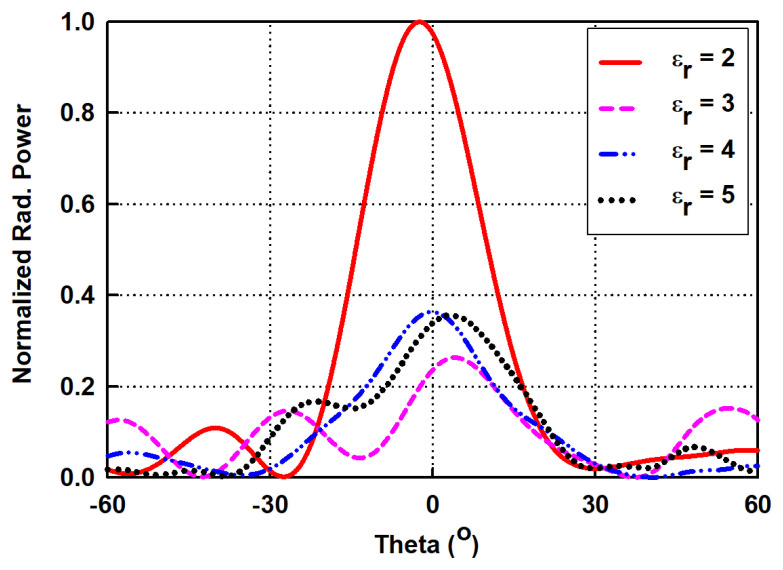
Normalized radiation power from the dielectric rod structure with varied relative permittivity, $\varepsilon_{r}$.

**Figure 9 sensors-21-04745-f009:**
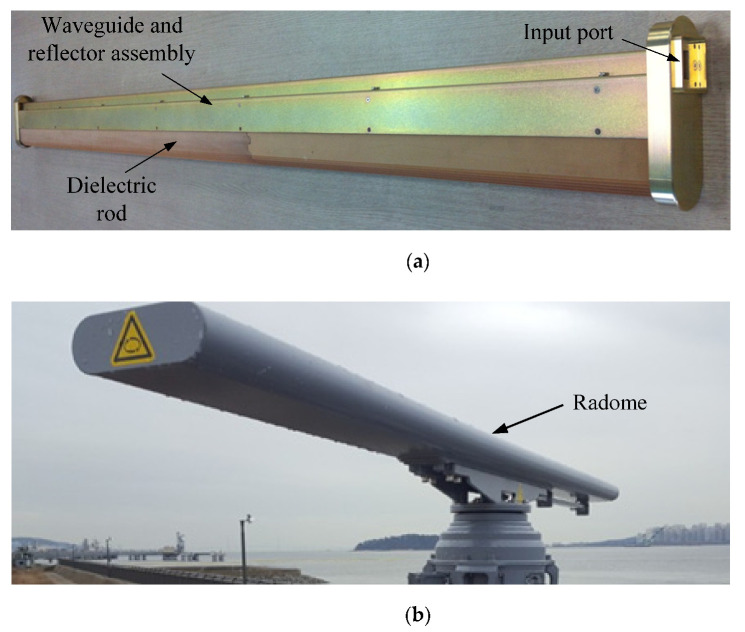
Photographs of the fabricated SWA array with a compact dielectric structure. (**a**) Assembled antenna prototype without a radome; (**b**) Outdoor proposed antenna installation in a radome.

**Figure 10 sensors-21-04745-f010:**
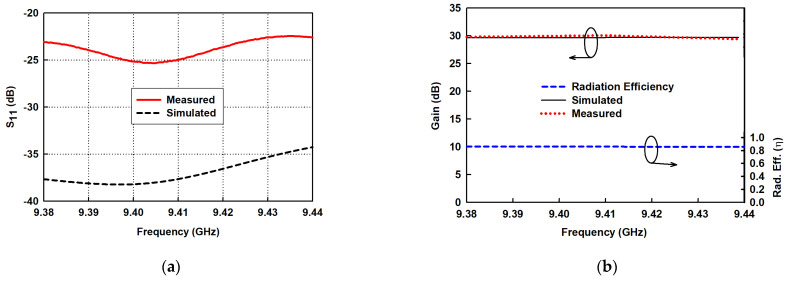
Measured and simulated results of proposed SWA array. (**a**) Reflection coefficients; (**b**) Gain curves and radiation efficiency.

**Figure 11 sensors-21-04745-f011:**
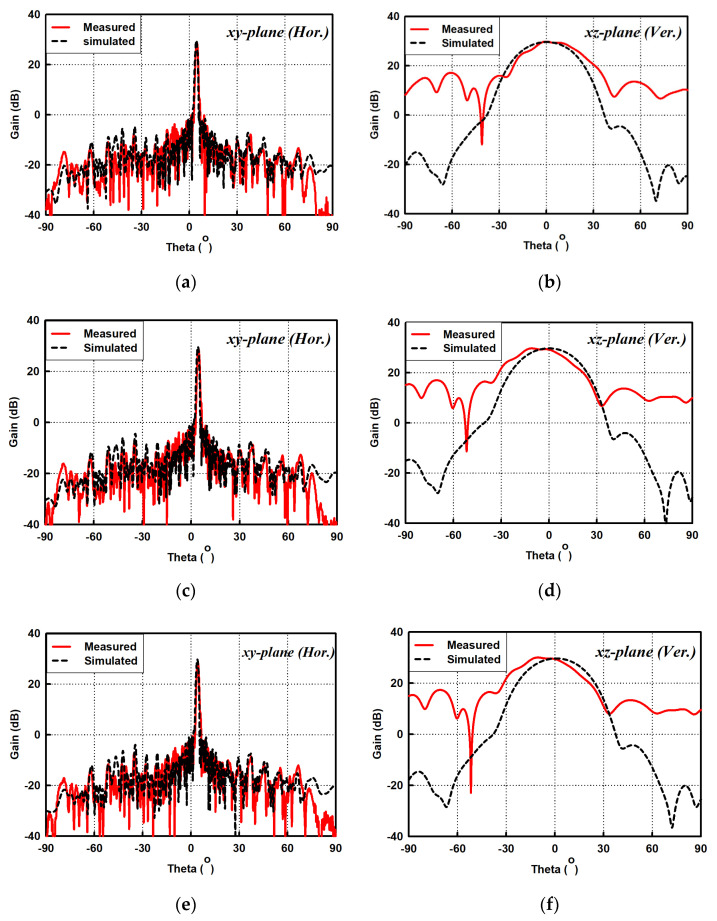
Measured and simulated radiation patterns of the proposed SWA array. (**a**) Horizontal (xy)-plane, 9.38 GHz; (**b**) Horizontal (xy)-plane, 9.41 GHz; (**c**) Horizontal (xy)-plane, 9.44 GHz; (**d**) Vertical (xz)-plane, 9.38 GHz; (**e**) Vertical (xz)-plane, 9.41 GHz; (**f**) Vertical (xz)-plane, 9.44 GHz.

**Figure 12 sensors-21-04745-f012:**
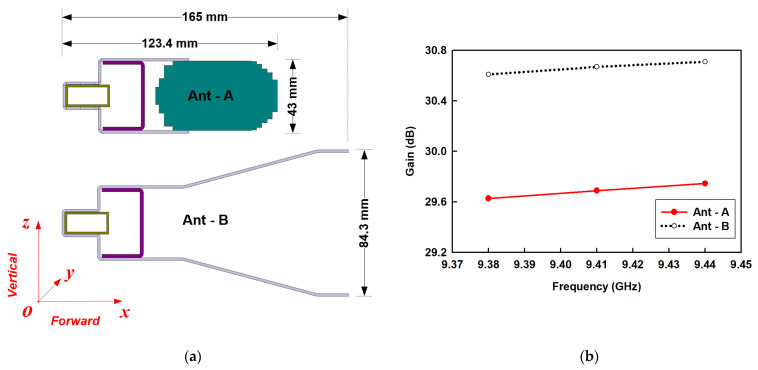
(**a**) Two dimensional analogues in xz (vertical) plane; (**b**) Gain results of the simulated conventional and proposed SWA arrays.

**Table 1 sensors-21-04745-t001:** Antenna design specifications for X-band marine radar.

Parameters	Value	Parameters	Value
Frequency	9.4 GHz ± 30 MHz	Polarization	Horizontal
Gain	≥28 dB	Beamwidth	Horizontal: 1.2° or less Vertical: 20° or higher
Voltage standing wave ratio (VSWR)	≤1.5	Sidelobelevel (SLL)	Horizontal: −10 dB or less Vertical: −10 dB or higher

**Table 2 sensors-21-04745-t002:** Comparison of the properties of different dielectric materials.

Material	A	B	C	D
Relative permittivity ($\varepsilon_{r}$)	2.1	2.53	1.7	1.53
Loss tangent (tanδ)	0.00015	0.0001	0.0001	0.00035
Density ($\mathrm{kg}/{\mathrm{cm}}^{3}$)	2.2	$1\times 10^{-3}$	$1\times 10^{-3}$	$8\times 10^{-5}$

**Table 3 sensors-21-04745-t003:** Comparison between the requirements and the measured data.

Parameters	Requirement	Numerical Data	Measured Data	
Frequency	9.4 GHz	9.4 GHz	9.4 GHz	ο
Gain	≥28 dB	≥29.62 dB	≥29.4 dB	ο
VSWR	≤1.5	≤1.5	≤1.5	ο
Polarization	Horizontal	Horizontal	Horizontal	ο
Beamwidth	Horizontal$\;\left(A_{z}\right)\leq 1.2{}^{\circ}$	≤1.2°	≤1.2°	ο
	Vertical$\;\left(E_{l}\right)\geq 20{}^{\circ}$	≥ 27.7°	≥27.8°	ο
SLL	Horizontal$\;\left(A_{z}\right)\leq -20$ dB	≤−20 dB	≤−20.8 dB	ο
	Vertical$\;\left(E_{l}\right)\leq -30$ dB	≤−31 dB	≤−12 dB	X

$A_{z}$ and $E_{l}$ denote azimuth and elevation planes, respectively.

## Data Availability

Not applicable.

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
