# Peer review of "Compact Slotted Waveguide Antenna Array Using Staircase Model of Tapered Dielectric-Inset Guide for Shipboard Marine Radar"

_sensors, 2021, doi:10.3390/s21144745_

Round 1
Reviewer 1 Report
see attached report

Author Response
Thank you for your valuable comments.We have done our best to ensure that the paper is amended on advice.
Please refer an attached.

Reviewer 2 Report
This is indeed a well written paper with clear demonstration of improvement of performance over the conventional designs. However, I find it quite surprising that there are no details provided on the design methodology. There is no basis presented behind how the optimised values have been obtained. There is also no mention of the simulation tool used for the study. Based on transmission line theory, there is a consolidated framework that explains the design and analysis of tapered slots that the authors have used. Inclusion of the discussion will make the results of the paper more reliable and backed by theory.Author Response
Thank you for your valuable comments.We have done our best to ensure that the paper is amended on advice. Please refer an attached.

Reviewer 3 Report
In this paper, a new proposal of slotted waveguide array antenna for shipboard marine radar applications in the X-band is presented. It essentially consists in substituting the traditional reflector by a tapered dielectric guide structure, which results in a significant size reduction, by maintaining the similar gain and side lobe level (the latter achieved in simulation but not in measurement). This reviewer finds that this work provides enough contribution to be published in the Sensors Journal. However, some minor changes should be addressed before being considered for publication:
- Please, check the reference numbers. It seems that some of them are wrong in the text. For example, in line 127 you might refer to [31] instead of [32]. Check also the rest of references in the document such as in line 68 and 73.
- Figure 2 should be Figure 1.
- The authors claim in line 80 that “this phenomenon inherently improves the sidelobe level”. Please, clarify this point or add a reference where the reader can find the explanation.
- Please, include in the text the name of the software used for the electromagnetic simulations.
- Please, clarify in the revised version if the proposed antenna has an absorber at the end. In such case, include the simulated and measured S21 parameter in the document and indicate the power dissipated in the absorber.
- The authors claim that the proposed structure achieves an improvement in the radiation efficiency up to 80%. Please, clarify if efficiency refers to the ratio between radiated and incident power or to the common definition in traveling wave antennas which accounts for the power dissipated in the antenna before the signal reaches the absorber.
- Please correct the Fraunhofer distance in line 186. It should be R=2D^2/lambda.
- The physical parameters described in line 203 are not showed in any figure. Please, indicate them in a figure in a similar way as it was done in Fig. 6 or include an explanation for clarification.
- It seems that in the first column in Table 2, frequency, gain and VSWR should be respectively replaced by relative permittivity, loss tangent and weight. Please, correct this point in the revised manuscript.
- It seems that there is an error in the third column of table 3. The parameter “Title 4” should be replaced by Measured data. Clarify this point in the revised version, please.
Author Response

(The authors gave the same response as above.)
